# Deep Plug-and-Play Clustering with Unknown Number of Clusters

**An Xiao**                                              *an.xiao@huawei.com*
*Huawei Noah's Ark Lab*

**Hanting Chen**                                    *chenhanting@huawei.com*
*Huawei Noah's Ark Lab*

**Tianyu Guo**                                         *tianyu.guo@huawei.com*
*Huawei Noah's Ark Lab*

**Qinghua Zhang**                             *a.zhangqinghua@huawei.com*
*Huawei Noah's Ark Lab*

**Yunhe Wang**[*]                                     *yunhe.wang@huawei.com*
*Huawei Noah's Ark Lab*

**Reviewed on OpenReview:** *https://openreview.net/forum?id=6rbcq0qacA*

## Abstract

Clustering is an essential task for the purpose that data points can be classified in an unsupervised manner. Most deep clustering algorithms are very effective when given the number of clusters $K$. However, when $K$ is unknown, finding the appropriate $K$ for these algorithms can be computationally expensive via model-selection criteria, and applying algorithms with an inaccurate $K$ can hardly achieve the state-of-the-art performance. This paper proposes a plug-and-play clustering module to automatically adjust the number of clusters, which can be easily embedded into existing deep parametric clustering methods. By analyzing the goal of clustering, a split-and-merge framework is introduced to reduce the intra-class diversity and increase the inter-class difference, which leverages the entropy between different clusters. Specifically, given an initial clustering number, clusters can be split into sub-clusters or merged into super-clusters and converge to a stable number of $K$ clusters at the end of training. Experiments on benchmark datasets demonstrate that the proposed method can achieve comparable performance with the state-of-the-art works without requiring the number of clusters.

## 1 Introduction

Clustering is a fundamental problem in unsupervised learning, widely applied to various applications in computer vision. The goal of clustering is to partition similar data into the same cluster and dissimilar data into different clusters. Since the late 20th century, quite a few seminal works have been proposed to address this problem, including K-means (MacQueen et al., 1967), DBSCAN (Ester et al., 1996), OPTICS (Ankerst et al., 1999), STING (Wang et al., 1997), *etc.* With the success of neural networks, deep clustering methods have become one of the mainstream choices in unsupervised classification since they can achieve much higher performance than the traditional methods, especially in computer vision field. Xie et al. (2016) propose deep-embedded clustering to learn the representations and clusters using deep networks simultaneously. Yang et al. (2017) utilize a deep network to learn a K-means-friendly space for the clustering task. Ghasedi Dizaji et al.

---

[*]Corresponding author.

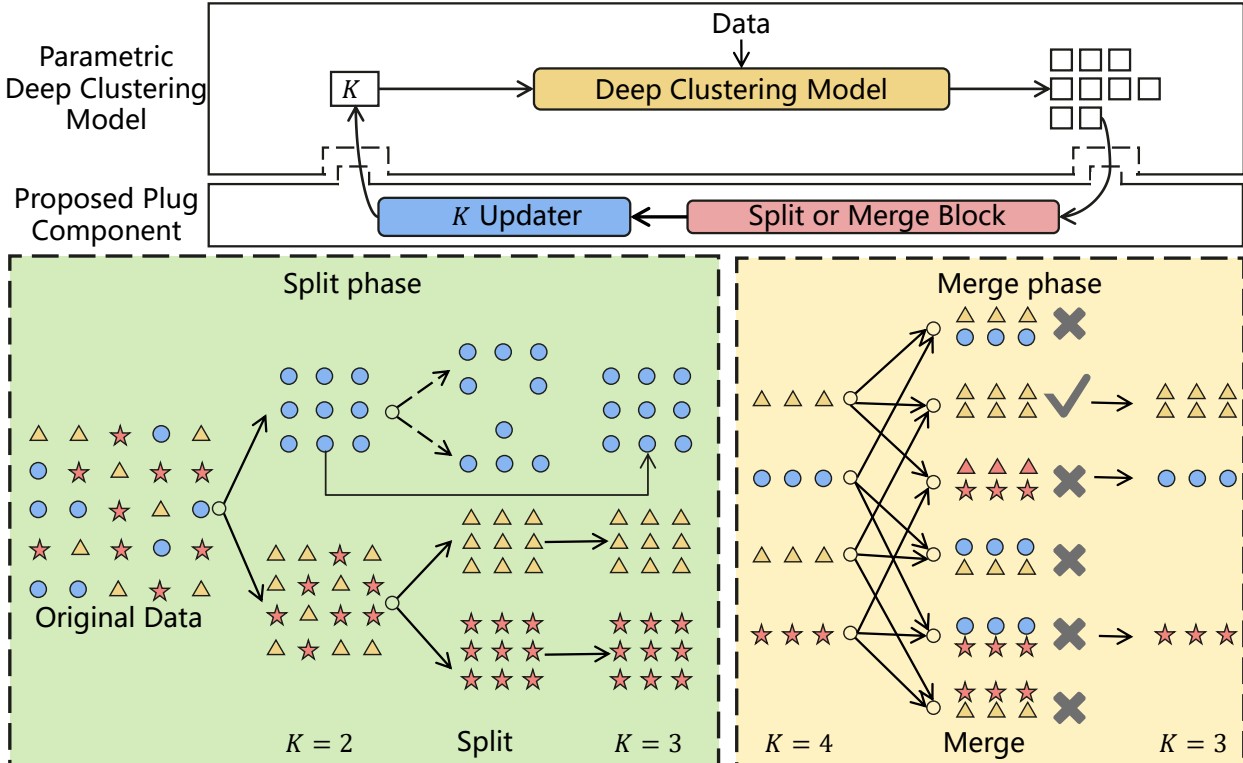

Figure 1: Proposed plug-and-play module can be applied in existing deep clustering methods. The split phase split the dissimilar data points in one clusters to minimize the inter-class diversity. The merge phase merge the similar data points in different clusters to maximize the intra-class diversity.

(2017) introduce a convolutional auto-encoder to joint learn the embedding and cluster assignments. Yang et al. (2016) utilize the variational auto-encoder to model the data generatively with a Gaussian Mixture Model.

Although the works above have achieved great success, an important issue has not been well addressed, *i.e.*, the vast majority of existing deep clustering algorithms require the providing number of clusters when applied to an unlabeled dataset. Furthermore, these deep methods usually cluster different data points by the fully-connected layer, which is optimized end-to-end and should be pre-defined before the training process. Therefore, a series of works focus on finding the appropriate clustering numbers for clustering. Antoniak (1974) proposes the Dirichlet Process to calculate the number of clusters in Gaussian Mixture Model. Rodriguez & Laio (2014) utilize the grid search to find the optimal number of clusters. However, applying these algorithms can be computationally expensive via model-selection criteria, especially for deep clustering methods whose training process is time-consuming. There exists few deep clustering works (Chen, 2015; Shah & Koltun, 2018; Wang et al., 2021; Zhao et al., 2019; Tapaswi et al., 2019; Ronen et al., 2022) trying to analyze the clustering task without knowing the number of clusters. For example, Tapaswi et al. (2019) introduce the ball cluster learning to carve the data space into balls for each cluster. Ronen et al. (2022) propose the DeepDPM by extending the Dirichlet Process Gaussian Mixture Model. However, the performance of these deep non-parametric methods (namely, methods that do not require the number of clusters $K$) are still far from aforementioned parametric methods, since it is difficult to train the clustering network and find the best $K$ simultaneously.

In this paper, we propose a deep plug-and-play clustering method with unknown number of clusters, which is illustrated in Figure 1. Instead of developing an effective non-parametric method, we aim to introduce a plug-and-play module that can be easily embedded into the state-of-the-art parametric clustering method without affecting its performance. Specifically, we analyze the goal of clustering in the non-parametric situation, that is to minimize the intra-class diversity and maximize the inter-class difference. A class split scheme is

introduced to split a cluster into sub-clusters if intra-class diversity is large. The clusters will be merged when the inter-class difference between clusters is small. We measure the similarity by leveraging the entropy between data points. The proposed algorithm can be applied to existing clustering methods and achieve state-of-the-art performance in various datasets without knowing the number of clusters. Experimental results demonstrate that the proposed split and merge scheme can successfully find the clustering numbers and train the clustering network.

## 2 Related Works

**Deep Clustering methods.** Deep learning has shown its great power in several fields, with no exception for the clustering method. From the very beginning, deep belief networks (DBN) were introduced to establish a deep clustering framework. Starting from this, AE (Bengio et al., 2006) extends DBN to process input with continuous value and proposes an unsupervised pre-train strategy to describe the similarity. DAE (Vincent et al., 2010) further enhance DBN with denoising as prior. With the development of neural networks, DeCNN (Zeiler et al., 2010) builds a deconvolution module to extract mid-level representations in an unsupervised way with a sparsity constraint, which achieves noise-robust clustering results. Several in-style ideas have been injected into the deep clustering methods in recent years. DDC (Chang et al., 2019) combines global and local constraints to investigate patterns' relationships and enhance the discriminative ability of features. CC (Li et al., 2021) proposes an online clustering method with the utilization of contrastive learning principles.

From the perspective of feature extraction and clustering, JULE (Yang et al., 2016) updates clusters and features alternatively in a recurrent process. Furthermore, DAC (Chang et al., 2017) bridges feature extraction with clustering by defining the clustering problem as a binary pairwise-classification framework. While SCAN (Van Gansbeke et al., 2020) disentangles the feature extraction step and the clustering step again, which prevents the model from being influenced by low-level features. Meanwhile, ADC (Haeusser et al., 2018) introduces centroid variables and optimizes them according to the objective training function, which allows no more additional classifiers to be trained separately.

In practice, the number of clusters $k$ is usually not easy to access. Based on this situation, several methods are proposed to estimate $k$ or avoid using it. Lewis et al. (2004) utilizes DPM sampler to get $k$. DeepDPM (Ronen et al., 2022) proposes a split or merge strategy and tries to remove the requirement of $k$.

**Non-parametric Classical Clustering.** A group of classical clustering algorithms is free of the prior of $k$. Density-based clustering algorithms are a representative family of clustering algorithms that do not require parameter $k$. DBSCAN (Ester et al., 1996) is based on density and does not require the prior of $k$. Later, Optics (Ankerst et al., 1999) is proposed to address the shortage of DBSCAN that makes it hard to fit the data of varying densities. Another well-known family of non-parametric clustering algorithms is the hierarchical clustering algorithm, such as BIRCH (Zhang et al., 1996), CHAMELEON (Karypis et al., 1999), ROCk (Guha et al., 2000).

## 3 Method

Denote $\{x_i\}_{i=1}^N$ as the whole $N$ data points, the goal for clustering is to partition these data points into different groups, where similar data should be put in the same group. There are various index to measure the performance of clustering results Halkidi et al. (2002), including Sum of Squared Error (SSE), Silhouette Coefficient, Calinski-Harabaz Index, Davies-Bouldin Index, etc. For the ease of analysis, we take two basic measurement for the evaluating the clustering results: Compactness and Separation. Given $K$ as the number of clusters, the Compactness $CP$ and Separation $SP$ can be formulated as:

$$CP = \sum_{k \in [K]} \sum_{i,j} d(\mathbf{x}_i^k, \mathbf{x}_j^k),$$

$$SP = \sum_{k_1,k_2 \in [K]}^{k_1 \neq k_2} \sum_{i,j} d(\mathbf{x}_i^{k_1}, \mathbf{x}_j^{k_2}),$$

(1)

where $\mathbf{x}_i^k$ denotes the $i$-th sample in the $k$-th cluster, $d(\cdot, \cdot)$ is a metric to evaluate the distance between two samples, and $[K]$ is the positive integer set from 1 to $K$, *i.e.* $\{1, 2, ..., K\}$. Compactness measures the distance between samples in the same class, while the Separation measures the distance between samples in different classes. The goal of clustering can be regarded to minimize Compactness and maximize Separation. Since the number of clusters $K$ is usually unknown in real-world applications, we also need to find the optimal value of $K^*$ for the clustering algorithm. For the simplicity of notation, we denote $D(k, k) = \sum_{i,j} d(\mathbf{x}_i^k, \mathbf{x}_j^k)$ as the inter-class similarity and $D(k_1, k_2) = \sum_{i,j} d(\mathbf{x}_i^{k_1}, \mathbf{x}_j^{k_2})$ as the intra-class similarity . The whole optimization problem to clustering with unknown number of clusters can be formulated as:

$$\arg\min_K \sum_{k \in [K]} D(k, k) - \frac{\lambda}{K} \sum_{\substack{k_1, k_2 \in [K]}}^{k_1 \neq k_2} D(k_1, k_2), \tag{2}$$

where $\lambda$ is a hyper-parameter to balance the two terms and is divided by $K$ to calculate the averaged distance between clusters. When utilizing traditional clustering algorithm, one can search the number of clusters $K$ in traversal since the computational cost is small. However, this traversal searching is unsuitable for deep learning based clustering algorithms since they are computationally expansive. To this end, we propose a plug-and-play clustering module to gradually adjust $K$ by a split and merge scheme during one training process for deep clustering methods.

### 3.1 Cluster Split to Minimize Inter-Class Diversity

We first consider the situation that current number of clusters $K'$ is smaller than the optimal value $K^*$. In this situation, the first term in the optimization problem 2 becomes the dominate term, since many dissimilar data points are classified into the same cluster due to the limited clustering numbers, though they are generally supposed to belong to different clusters. Therefore, a split scheme to increase the number of clusters is required.

As it is difficult to directly estimate the optimal value $K^*$, we propose to increase $K$ in a greedy manner. Given a clustering number $K'$, a parametric clustering method can classify the data points into $K'$ classes. We aim to investigate whether the points in each cluster can be further divided into two clusters. We propose a hierarchical clustering algorithm to solve this problem.

As mentioned above, we optimize the clustering network by minimizing the intra-class similarity between the data points. Hence, if data points in one cluster can be easily divided into two sub-clusters, the points in this cluster are dissimilar and ought to be further separated. We then determine whether each cluster should be split by analyzing optimization problem 2.

**Proposition 1.** *When sub-clusters $K^1$ and $K^2$ satisfy the following equation:*

$$\left( \frac{2\lambda}{K+1} + 2 \right) \mathrm{D}(K^1, K^2) \geq \frac{\lambda}{K(K+1)} \sum_{\substack{k_1, k_2 \in [K]}}^{k_1 \neq k_2} \mathrm{D}(k_1, k_2), \tag{3}$$

*applying the split phase will decrease the value of optimization problem 2.*

In order to determine whether a cluster should be split, similarity between the data points in two sub-clusters needs to be evaluated. There are various metrics to measure the similarity between distributions, including mutual information, Kullback-Leibler (KL) divergence, Jensen-Shannon (JS) divergence and Wasserstein distance. We use the JS divergence to measure the similarity between two sub-clusters, since it is symmetry

and easy to calculate. Specifically, the JS divergence can be formulated as:

$$
\begin{aligned}
\mathrm{JS}(K^1\|K^2) =& \frac{1}{2}D_{\mathrm{KL}}(K^1\|\frac{K^1+K^2}{2}) + \frac{1}{2}D_{\mathrm{KL}}(K^2\|\frac{K^1+K^2}{2}) \\
=& \sum_i P_{K^1}(\mathbf{x}_i^{K^1})\log(\frac{2P_{K^1}(\mathbf{x}_i^{K^1})}{P_{K^1}(\mathbf{x}_i^{K^1})+P_{K^2}(\mathbf{x}_i^{K^2})}) + \sum_i P_{K^2}(\mathbf{x}_i^{K^2})\log(\frac{2P_{K^2}(\mathbf{x}_i^{K^2})}{P_{K^1}(\mathbf{x}_i^{K^1})+P_{K^2}(\mathbf{x}_i^{K^2})}) \\
=& \sum_i \left[ P_{K^1}(\mathbf{x}_i^{K^1})\log(P_{K^1}(\mathbf{x}_i^{K^1})) + P_{K^2}(\mathbf{x}_i^{K^2})\log(P_{K^2}(\mathbf{x}_i^{K^2})) \right] \\
& - \sum_i \left[ P_{K^1}(\mathbf{x}_i^{K^1}) + P_{K^2}(\mathbf{x}_i^{K^2}) \right] \log(\frac{P_{K^1}(\mathbf{x}_i^{K^1})+P_{K^2}(\mathbf{x}_i^{K^2})}{2})
\end{aligned}
\tag{4}
$$

where $P_{K^1}(\cdot)$ or $P_{K^2}(\cdot)$ denote the probability of data points classified into cluster $K^1$ or $K^2$. By introducing the JS divergence, we can evaluate whether the two sub-clusters are dissimilar. When the JS divergence becomes larger, two sub-clusters can be divided more easily. Therefore, we set a threshold $T_s$ for splitting the clusters. If the JS divergence of the two sub-clusters $\mathrm{JS}(K^1\|K^2) \geq T_s$, the split operation is employed in the main clustering stage for the $k$-th cluster. Specifically, the weights for $k$-th cluster will be copycatted for another cluster and the number of clusters is therefore increased. In fact, the threshold can be determined by inequation 3 by utilizing the JS divergence as the distance measurement:

$$
\begin{aligned}
\left( \frac{2\lambda}{K+1} + 2 \right) \mathrm{JS}(K^1\|K^2) &\geq \frac{\lambda}{K(K+1)} \sum_{k_1,k_2 \in [K]} \mathrm{JS}(k_1\|k_2) \\
\mathrm{JS}(K^1\|K^2) &\geq \frac{\lambda}{2K(\lambda+K+1)} \sum_{k_1,k_2 \in [K]} \mathrm{JS}(k_1\|k_2).
\end{aligned}
\tag{5}
$$

Therefore, we can set the threshold for the determination of split according to following equation:

$$
T_s = \frac{\lambda}{2K(\lambda+K+1)} \sum_{k_1,k_2 \in [K]} \mathrm{JS}(k_1\|k_2).
\tag{6}
$$

We split the data points in each cluster $k$ into two sub-clusters $k^1$ and $k^2$ utilizing the algorithm $\mathcal{A}$. More concretely, we directly add $K'$ assistant 2-class classification feed-forward neural networks to the $K'$ dimensional output of algorithm $\mathcal{A}$, and the overall number of clusters is expanded to $2K'$. To make the sub-clusters different after split, we introduce a split loss for separating the newly added cluster with the original cluster by maximizing the JS divergence between the two sub-clusters:

$$
L_{split} = L_{ori} - \gamma \sum_{i,j} \alpha_{ij} \mathrm{JS}(k_i\|k_j),
\tag{7}
$$

where $L_{ori}$ denotes the original training loss of algorithm $\mathcal{A}$. We set $\alpha_{ij} = 1$ if $k_i$ and $k_j$ are the split sub-clusters, otherwise $\alpha_{ij} = 0$. By hierarchically dividing the sub-clusters and introducing the JS divergence, we can split the dissimilar data in one cluster into sub-clusters, therefore increasing the number of clusters.

## 3.2 Cluster Merge to Maximize Intra-Class Diversity

Another important assumption in clustering is that the data point in different clusters should be dissimilar. When the number of clusters is much higher than its optimal value, the similar data point could be divided into different clusters, which reduces the inter-class diversity of the clustering algorithm. Consequently, we also propose a class merge scheme to decrease the number of clusters and merge the similar data points in different clusters into the one cluster.

Same as the problems mentioned in split scheme, it is difficult and unstable to directly merge all the similar clusters for deep clustering methods. Thus, we apply a greedy manner to merge two clusters at one time. The JS divergence can measure the similarity of the data points between two clusters. Denote $k$ as the

---

**Algorithm 1** Deep plug-and-play clustering with unknown number of clusters.

---

**Require:** A parametric clustering algorithm $\mathcal{A}$, an initial number of clusters $K'$, the split and merge hyperparameter $\lambda$, and the dataset $\mathcal{X}$ to be clustered.
1: Initialize the clustering network $\mathcal{N}$, set current number of cluster $K^* = K'$;
2: **repeat**
3:     Apply the algorithm $\mathcal{A}$ to training the network $\mathcal{N}$ with current number of cluster $K^*$;
4:     **Phase 1: Split clusters.**
5:     **for** $i$ in $[K^*]$ **do**
6:         Split data $\mathcal{X}_i$ of $i$-th cluster into two sub-clusters $K^1$ and $K^2$ using algorithm $\mathcal{A}$;
7:         Calculate the $\text{JS}(k^1 \| k^2)$ and the threshold $T_s$ according to Equ. 6;
8:         **if** $\text{JS}(K^1 \| K^2) > T_s$ **then**
9:             Split and increase $K^*$;
10:         **end if**
11:     **end for**
12:     Apply the algorithm $\mathcal{A}$ to training the network $\mathcal{N}$ with $K^*$ and Equ. 7;
13:     **Phase 2: Merge clusters.**
14:     Find candidate merge clusters $K^i$ and $K^j$ according to Equ. 8;
15:     Calculate the $\text{JS}(K^i \| K^j)$ and the threshold $T_m$ according to Equ. 12;
16:     **if** $\text{JS}(K^i \| K^j) < T_m$ **then**
17:         Merge $K^i$ and $K^j$, decrease $K^*$ ;
18:     **end if**
19:     Apply the algorithm $\mathcal{A}$ to training the network $\mathcal{N}$ with $K^*$;
20: **until** convergence

**Ensure:** A well-trained clustering network $\mathcal{N}$ with an accurately estimated number of clusters $K^*$.

---

current number of clusters, we aim to find the most similar clusters $K^1$ and $K^2$, which can be formulated as:

$$K^1, K^2 = \underset{k_1, k_2 \in [K], k_1 \neq k_2}{\arg\min} \text{JS}(k_1 \| k_2). \tag{8}$$

By traversing the data points in all clusters and calculating the JS divergence pariwisely for each two clusters, we select two clusters with the largest JS divergence. Likewise, we set a threshold $T_m$ to determine whether we should merge the clusters. If the JS divergence of these two clusters meets $\text{JS}(K^1 \| K^2) \leq T_m$, they will be merged into one cluster. Specifically, the weights in the fully connected layer for the two clusters will be averaged to obtain the weight of the merged cluster. Without loss of generalization, we can assume that the two clusters $K^1$ and $K^2$ are the last two clusters.

**Proposition 2.** *When two clusters $K^1$ and $K^2$ satisfy the following equation:*

$$\left( \frac{2\lambda}{K} + 2 \right) \text{D}(K^1, K^2) \leq \frac{\lambda}{K(K-1)} \sum_{\substack{k_1, k_2 \in \{[K-2], K^m\} }}^{k_1 \neq k_2} \text{D}(k_1, k_2). \tag{9}$$

*applying the merge phase will decrease the value of optimization problem 2, where $K^m$ denotes the merged cluster.*

By utilizing the JS divergence as the distance measurement, we have:

$$\left( \frac{2\lambda}{K} + 2 \right) \text{JS}(K^1 \| K^2) \leq \frac{\lambda}{K(K-1)} \sum_{\substack{k_1, k_2 \in \{[K-2], K^m\} }}^{k_1 \neq k_2} \text{JS}(k_1 \| k_2). \tag{10}$$

Finally, the relationship based on JS divergence can be formulated as follows,

$$\text{JS}(K^1 \| K^2) \leq \frac{\lambda}{2(K-1)(\lambda+K)} \sum_{\substack{k_1, k_2 \in \{[K-2], K^m\} }}^{k_1 \neq k_2} \text{JS}(k_1 \| k_2). \tag{11}$$

Then we set the threshold for the determination of merge according to following equation,

$$T_m = \frac{\lambda}{2(K-1)(\lambda+K)} \sum_{\substack{k_1, k_2 \in \{[K-2], K^m\} }}^{k_1 \neq k_2} \mathrm{JS}(k_1 \| k_2). \tag{12}$$

Same as the strategy of the split phase, if the JS divergence between two merged clusters $\mathrm{JS}(K^1 \| K^2)$ is smaller than the threshold obtained above, these two clusters will be decided to be merged.

### 3.3 Split-Merge Scheme to Estimate the Number of Clusters

The overall algorithm of the proposed method can be found in Algorithm 1. First, the network is trained by a parametric clustering algorithm $\mathcal{A}$ with an initial number of clusters $K'$. When $K'$ is far from the optimal value $K^*$, the performance of the network will suffer degradation. In order to sovle the problem, we propose to iteratively apply the split and merge module to estimate the number of clusters automatically alongside the training process. Concretely, the training process of our method is as following, we first train the whole network till the current loss does not decrease anymore. Then we compute the value of current $T_m$ and $T_s$ for merge or split, separately. When the network training is converged at the current $K^*$ clusters, the threshold $T_s$ will be larger than $T_m$. By measuring the distances of samples between every two clusters and comparing to $T_m$, we will start merge operation if the condition is satisfied. Once merge is executed, we will pass the split phase and directly jump into the next loop. The situation is the same for split, once split condition is satisfied, merge phased is passed. And at the end of whole training, the inferred $K$ converges when JS divergence is between $T_m$ and $T_s$. The proposed module can be applied in various parametric clustering algorithms to achieve a accurate estimation of the number of clusters and the performances are comparable to the original clustering algorithms.

**Lemma 3.** *The split and merge condition 3 and 9 cannot be satisfied simultaneously.*

**Lemma 4.** *The number of clusters can only be monotonically changed during the training process.*

*Proof.* We denote $V(K)$ as $\sum_{k \in [K]} D(k, k) - \frac{\lambda}{K} \sum_{k_1, k_2 \in [K]}^{k_1 \neq k_2} D(k_1, k_2)$. From Proposition 1, we conclude that the split condition 3 is equal to:

$$\sum_{k \in \{[K-1], K^1, K^2\}} \mathrm{D}(k, k) - \frac{\lambda}{K+1} \sum_{\substack{k_1, k_2 \in \{[K-1], K^1, K^2\}}}^{k_1 \neq k_2} \mathrm{D}(k_1, k_2) \leq \sum_{k \in [K]} \mathrm{D}(k, k) - \frac{\lambda}{K} \sum_{k_1, k_2 \in [K]} \mathrm{D}(k_1, k_2), \tag{13}$$

which can be formulated as $V(K+1) < V(K)$. From the proof of Proposition 2, we can also conclude that the merge condition 9 is equal to $V(K-1) < V(K)$. From the proof of Lemma 3, we conclude that there is impossible that $V(K-1) < V(K)$ and $V(K+1) < V(K)$. Therefore, it remains the probability that $V(K+1) < V(K) < V(K-1)$, $V(K+1) > V(K) > V(K-1)$ and $V(K+1) > V(K) < V(K-1)$. If $V(K+1) > V(K) < V(K-1)$, the algorithm converged. If $V(K+1) < V(K) < V(K-1)$ meets the split condition and the number of clusters $K$ grows to $K+1$, the merge condition cannot be satisfied since $V(K+1) < V(K)$, which means the number of clusters $K+1$ cannot go back $K$. The proof is same when meets the merge condition. Therefore, the number of clusters can only be monotonically changed. $\square$

**Proposition 5 (Convergence of the proposed algorithm).** *The split-merge scheme using the inequation 3 and 9 will converge at an $O(\frac{1}{2^n})$ speed, where $n$ is the number of acting the split-merge schemes. Moreover, the scheme will recover the ground truth number of clusters $K^*$ when*

$$K^* = \arg\min_K \sum_{k \in [K]} D(k, k) - \frac{\lambda}{K} \sum_{\substack{k_1, k_2 \in [K]}}^{k_1 \neq k_2} D(k_1, k_2), \tag{14}$$

From Lemma 4, since the the number of clusters can only be monotonically changed and bounded ($K \geq 1$ and $K \leq m$ where $m$ is the number of samples), the proposed algorithm can finally converge. Moreover, we conclude that the algorithm converged when $V(K^*+1) > V(K^*) < V(K^*-1)$, which means 14 is satisfied.

The complete proof of the above lemmas and proposition can be found in the appendix.

Table 1: Comparison with state-of-the-art methods. $K_0$ denotes the initial number of clusters.

| Method | CIFAR-10 | | | CIFAR-100 | | | STL-10 | | | ImageNet-10 | | |
|---|---|---|---|---|---|---|---|---|---|---|---|---|
| Metric | NMI | ACC | ARI | NMI | ACC | ARI | NMI | ACC | ARI | NMI | ACC | ARI |
| AC (Gowda & Krishna, 1978) | 10.5 | 22.8 | 6.5 | 9.8 | 13.8 | 3.4 | 23.9 | 33.2 | 14.0 | 13.8 | 24.2 | 6.7 |
| NMF (Cai et al., 2009) | 8.1 | 19.0 | 3.4 | 7.9 | 11.8 | 2.6 | 9.6 | 18.0 | 4.6 | 13.2 | 23.0 | 6.5 |
| AE (Bengio et al., 2006) | 23.9 | 31.4 | 16.9 | 10.0 | 16.5 | 4.8 | 25.0 | 30.3 | 16.1 | 21.0 | 31.7 | 15.2 |
| DAE (Vincent et al., 2010) | 25.1 | 29.7 | 16.3 | 11.1 | 15.1 | 4.6 | 22.4 | 30.2 | 15.2 | 20.6 | 30.4 | 13.8 |
| DCGAN (Radford et al., 2015) | 26.5 | 31.5 | 17.6 | 12.0 | 15.1 | 4.5 | 21.0 | 29.8 | 13.9 | 22.5 | 34.6 | 15.7 |
| DeCNN (Zeiler et al., 2010) | 24.0 | 28.2 | 17.4 | 9.2 | 13.3 | 3.8 | 22.7 | 29.9 | 16.2 | 18.6 | 31.3 | 14.2 |
| VAE (Kingma & Welling, 2013) | 24.5 | 29.1 | 16.7 | 10.8 | 15.2 | 4.0 | 20.0 | 28.2 | 14.6 | 19.3 | 33.4 | 16.8 |
| JULE (Yang et al., 2016) | 19.2 | 27.2 | 13.8 | 10.3 | 13.7 | 3.3 | 18.2 | 27.7 | 16.4 | 17.5 | 30.0 | 13.8 |
| DEC (Xie et al., 2016) | 25.7 | 30.1 | 16.1 | 13.6 | 18.5 | 5.0 | 27.6 | 35.9 | 18.6 | 28.2 | 38.1 | 20.3 |
| DAC (Chang et al., 2017) | 39.6 | 52.2 | 30.6 | 18.5 | 23.8 | 8.8 | 36.6 | 47.0 | 25.7 | 39.4 | 52.7 | 30.2 |
| ADC (Haeusser et al., 2018) | - | 32.5 | - | - | 16.0 | - | - | 53.0 | - | - | - | - |
| DDC (Chang et al., 2019) | 42.4 | 52.4 | 32.9 | - | - | - | 37.1 | 48.9 | 26.7 | 43.3 | 57.7 | 34.5 |
| DCCM (Wu et al., 2019) | 49.6 | 62.3 | 40.8 | 28.5 | 32.7 | 17.3 | 37.6 | 48.2 | 26.2 | 60.8 | 71.0 | 55.5 |
| IIC (Ji et al., 2019) | 51.3 | 61.7 | 41.1 | - | 25.7 | - | 43.1 | 49.9 | 29.5 | - | - | - |
| MMDC (Shiran & Weinshall, 2021) | 57.2 | 70.0 | - | 25.9 | 31.2 | - | 49.8 | 61.1 | - | 71.9 | 81.1 | - |
| PICA (Huang et al., 2020) | 56.1 | 64.5 | 46.7 | 29.6 | 32.2 | 15.9 | - | - | - | 78.2 | 85.0 | 73.3 |
| DCCS (Zhao et al., 2020) | 56.9 | 65.6 | 46.9 | - | - | - | 37.6 | 48.2 | 26.2 | 60.8 | 71.0 | 55.5 |
| DHOG (Darlow & Storkey, 2020) | 58.5 | 66.6 | 49.2 | 25.8 | 26.1 | 11.8 | 41.3 | 48.3 | 27.2 | - | - | - |
| GATCluster (Niu et al., 2020) | 47.5 | 61.0 | 40.2 | 21.5 | 28.1 | 11.6 | 44.6 | 58.3 | 36.3 | 59.4 | 73.9 | 55.2 |
| IDFD (Tao et al., 2021) | 71.4 | 81.5 | 66.3 | 42.6 | 42.5 | 26.4 | 64.3 | 75.6 | 57.5 | 89.8 | 95.4 | 90.1 |
| CC (Li et al., 2021) | 70.5 | 79.0 | 63.7 | 43.1 | 42.9 | 26.6 | 76.4 | 85.0 | 72.6 | 85.9 | 89.3 | 82.2 |
| MiCE (Tsai et al., 2020) | 73.7 | 83.5 | 69.8 | 43.6 | 44.0 | 28.0 | 63.5 | 75.2 | 57.5 | - | - | - |
| SCAN (Van Gansbeke et al., 2020) | 71.2 | 81.8 | 66.5 | 44.1 | 42.2 | 26.7 | 65.4 | 75.5 | 59.0 | 86.2 | 92.0 | 83.3 |
| Ours ($K_0$=3) | 71.9 | 82.4 | 67.5 | 45.2 | 43.8 | 28.1 | 64.3 | 74.5 | 57.6 | 88.6 | 91.2 | 87.1 |
| Ours ($K_0$=20) | 71.1 | 81.6 | 66.2 | - | - | - | 65.1 | 74.7 | 58.9 | 86.9 | 91.8 | 84.7 |
| Ours ($K_0$=30) | - | - | - | 44.9 | 43.1 | 27.8 | - | - | - | - | - | - |

## 4 Experiments

In this section, we evaluate the effectiveness of the proposed method by comparison with the state-of-the-arts and ablation studies.

**Datasets and Metrics.** We conduct experimental evaluation on the CIFAR-10 (Krizhevsky et al., 2009), CIFAR100-20 (Krizhevsky et al., 2009), STL-10 (Coates et al., 2011), ImageNet-10 (Deng et al., 2009) and ImageNet-50 (Deng et al., 2009) datasets. We train and evaluate these datasets using their train and validation split to study the generalization properties of the proposed method. To evaluate the clustering performance of various methods, we adopt three standard evaluation metrics: Clustering Accuracy (ACC), Normalized Mutual Information (NMI) and Adjusted Rand Index (ARI).

**Training details.** We apply the proposed plug-and-play module in the deep clustering method SCAN (Van Gansbeke et al., 2020) without changing their original training settings, including the learning rate, weight decay, neural network architecture, data pre-processing and augmentation. For the proposed method, the hyper-parameters $\lambda$ is set as 2.0.

### 4.1 Comparison with the State-of-the-art

To demonstrate the effectiveness of the proposed method, we report the performance with the widely used deep clustering method SCAN (Van Gansbeke et al., 2020). Table 1 shows the results on the CIFAR-10, CIFAR100-20, STL-10 and ImageNet-10 datasets. As a result, traditional clustering algorithms such as AC and NMF can hardly achieve good performance on these datasets, which results in ∼20% accuracies. Deep learning-based clustering methods, such as DDC, DAC and DCCM, achieve much better performances. However, the state-of-the-art clustering algorithms (*e.g.*, CC, MiCE and SCAN) are parametric, which requires the exact number of clusters $K$. Therefore, we propose a plug-and-play module to solve this problem. We apply the proposed module to the SCAN algorithm. For CIFAR-10, STL-10 and ImageNet-10,

Table 2: Comparison of the mean inferred value for $K$ for different initiations.

| Dataset | Initial $K$ | | |
|---|---|---|---|
| | 3 | 20 | 30 |
| CIFAR-10 | 10±0 | 10±0 | - |
| CIFAR100-20 | 19.7±2.1 | - | 19.8±1.5 |
| STL-10 | 9.7±0.5 | 10.3±0.9 | - |
| ImageNET-10 | 10±0 | 10.3±0.5 | - |

Table 3: Comparison with state-of-the-art methods (3 runs) on the ImageNet-50 dataset.

| | Init k | Inferred k | ACC | MNI | ARI |
|---|---|---|---|---|---|
| SCAN Van Gansbeke et al. (2020) | 50 | - | 73.7±1.7 | 79.7±0.6 | 61.8±1.3 |
| SCAN Van Gansbeke et al. (2020) | 10 | - | 19.4±0.1 | 60.6±0.4 | 23.0±0.3 |
| DBSCAN Ester et al. (1996) | - | 16.0 | 24.0±0.0 | 52.0±0.0 | 9.0±0.0 |
| moVB Hughes & Sudderth (2013) | - | 46.2±1.3 | 55.0±2.0 | 70.0±1.0 | 43.0±1.0 |
| DPM Sampler Dinari et al. (2019) | - | 72.0±2.6 | 57.0±1.0 | 72.0±0.0 | 43.0±1.0 |
| DeepDPM Ronen et al. (2022) | 10 | 55.3±1.5 | 66.0±1.0 | 77.0±0.0 | 54.0±1.0 |
| Ours | 10 | 50.6±1.7 | 73.3±1.3 | 80.1±0.7 | 62.3±1.5 |

we set the initial numbers of clusters $K_0$ to be 3 and 20, which covers both scenarios when the initial $K_0$ is less or greater than the ground-truth value of the number of clusters. Likewise, we set $K_0$ for CIFAR100-20 to be 3 and 30. As is shown in Table 1, our proposed method still achieves comparable performance with these parametric methods no matter whether the initial cluster number is small or large. Besides, as our method can automatically adjust the number of clusters, we report the inferred number of clusters in Table 2. The results demonstrate that inferred $K$ for our split-merge module at the end of training is close to the real number of clusters $K$ with different initiations, suggesting that our method effectively adjusts the number of clusters during training. Note that all results are averaged from 3 different runs.

We further evaluate the proposed method on the ImageNet-50 dataset, which is more complicated since 50 categories of images need to be clustered. We compare the proposed method with both parametric and non-parametric algorithms. As shown in Table 3, the parametric clustering algorithm (SCAN) achieves the best performance (73.7% accuracy) when given the actual number of clusters. However, in real-world scenarios, the number of clusters is usually unknown, and thus these parametric algorithms can hardly be applied (*e.g.*, When k=10, the accuracy of SCAN degrades to 19.4). Although the non-parametric clustering methods do not require the number of clusters, currently, the state-of-the-art method (DeepDPM) can only achieve a 66.0% accuracy, 77.0% NMI and 54.0% ARI on this dataset, which is much lower than parametric algorithms. The proposed split-merge module can address the aforementioned problem and achieve the best performance without the exact number of clusters. Moreover, the number of clusters estimated by ours is 50.6, which is ideally close to the ground ruth (50) and better than the estimated value of DeepDPM (55.3), further indicating the effectiveness of the proposed method in more complicated scenarios.

## 4.2 Generalizability of the Proposed Module

**On different deep clustering frameworks.** We combine our proposed split-merge module with another two paramatric deep clustering methods: NNM (Dang et al., 2021) and GCC (Zhong et al., 2021). Table 4 shows the clustering performances on CIFAR-10 with different initial $K$. Results show that for various parametric methods, when $K$ diverges form the ground-truth $K$, the performances deteriorate to a large extend. However, when embedded with our proposed module, those methods acheive much better clustering accuracies (*e.g.,*, when $K$=3, the accuracy of GCC increase from 29.0% to 84.1%) and yield an accurate $K$ close to the GT.

**On different domains of data.** We

Table 4: Performance of the proposed module with different clustering methods.

| Initial $K$ | Method | ACC | NMI | ARI | Inferred $K$ |
|---|---|---|---|---|---|
| 3 | NNM | 28.9±0.3 | 45.2±0.4 | 25.8±0.1 | - |
|  | NNM+Ours | 80.5±1.2 | 75.1±1.1 | 69.3±0.7 | 9.3 ±0.5 |
|  | GCC | 29.0±0.1 | 45.2±0.4 | 27.9±0.1 | - |
|  | GCC+Ours | 84.1±2.6 | 78.0±1.6 | 74.7±2.1 | 10.7 ±0.5 |
| 10 | NNM | 84.1±0.1 | 74.8±0.1 | 70.2±0.2 | - |
|  | NNM+Ours | 84.3±0.1 | 75.0±0.1 | 70.8±0.1 | 10.0±0.0 |
|  | GCC | 85.6±0.1 | 76.7±0.2 | 72.8±0.1 | - |
|  | GCC+Ours | 85.8±0.8 | 77.6±0.5 | 73.6±1.5 | 10.0±0.0 |
| 20 | NNM | 50.2±1.9 | 63.6±1.6 | 44.8±0.3 | - |
|  | NNM+Ours | 80.4±1.3 | 74.1±1.2 | 68.7±2.4 | 10.7±0.5 |
|  | GCC | 49.8±0.2 | 68.2±0.2 | 49.2±0.3 | - |
|  | GCC+Ours | 85.2±0.6 | 77.4±0.3 | 72.4±1.2 | 10.0±0.0 |

Table 5: Results on the CIFAR-10 dataset using different numbers of clusters $K_0$.

| $K_0$ | 3 | 10 | 15 | 20 | 30 | 40 | 50 | 100 |
|---|---|---|---|---|---|---|---|---|
| SCAN | 28.6 | 82.2 | 58.8 | 45.9 | 33.4 | 25.9 | 21.9 | 11.9 |
| Ours | 82.4 | 82.4 | 82.9 | 81.6 | 82.0 | 77.5 | 70.9 | 72.4 |
| $K_{\text{Inferred}}$ | 10 | 10 | 10 | 10 | 10 | 11 | 12 | 12 |

### 4.3 Ablation Studies

In this subsection, we further conduct ablation studies, including various initial clustering numbers $K_0$ and different components to analyze the proposed method.

**Various initial clustering numbers** $K_0$. As is shown in Algorithm 1, the proposed method needs an initial number of clusters $K_0$ to train the clustering network, then the split and merge module can be applied. Therefore, it is essential to investigate the influence of $K_0$ in the proposed method. Table 5 shows the models starting with different numbers of clusters. We compare the proposed method with a parametric clustering algorithm: SCAN.

Obviously, the performance of SCAN is sensitive to the number of clusters since it cannot adjust cluster numbers automatically during the training process. When the ground-truth number of clusters $K^*$ is known ($K_0 = K^* = 10$), SCAN can achieve the best accuracy (82.2%). However, when there is a gap between $K_0$ and $K^*$, the performance of SCAN degrades a lot. For example, when $K_0$=3 and $K_0$=100, the accuracies of SCAN are only 28.6% and 11.9%, which are far from its optimal accuracy (82.2%). In contrast, when utilizing the split and merge module, the proposed method achieves a stable ∼82% accuracy ($K_0$ from 3 to 30), which outperforms the original SCAN by a large margin. Even if the initial number is nowhere near the ground-truth clusters ($K_0$ from 40 to 100), the proposed method can still achieve accuracies above 70.0% and generate cluster numbers close to 10.

**Various values of hyper-parameter** $\lambda$. As is mentioned in the methods, the proposed threshold formulas in split or merge phase involve a pre-defined $\lambda$, which is an oracle hyper-parameter and cannot be known in advance. However, we clarify here that the value of $\lambda$ does not affect the optimization problem, as it is a fixed hyper-parameter that remains constant during the optimization process. We conduct an experiment to study the impact of different values of $\lambda$ in Table 6. Results show that when $\lambda$ is within a reasonable scope, the clustering performances remain steady. Besides, $\lambda$ is robust for different datasets with different number of clusters. We set $\lambda = 2$ for all datasets, which avoids the heavy hyper-parameters' tuning works.

**Different components of the proposed method.** To evaluate the effectiveness of each component in the proposed method, we conduct an ablation study in Table 7. We first study the case that the initial number of

Table 6: Results of the proposed method with different values of $\lambda$.

| $\lambda$ | 1.5 | 1.8 | 2 | 2.2 | 2.50 |
|---|---|---|---|---|---|
| CIFAR-20 | 39.84 | 43.57 | 44.71 | 41.34 | 41.40 |
| STL-10 | 69.46 | 71.14 | 78.40 | 76.49 | 75.53 |

Table 7: Performance of the proposed method under different ablations.

| | $K_0=3$ | | | $K_0=10$ | | | $K_0=20$ | | |
|---|---|---|---|---|---|---|---|---|---|
| | ACC | NMI | ARI | ACC | NMI | ARI | ACC | NMI | ARI |
| No split/merge | 28.7 | 46.2 | 26.4 | 82.2 | 71.6 | 66.8 | 47.3 | 62.3 | 43.2 |
| No split | 28.9 | 46.1 | 26.4 | 75.4 | 68.9 | 62.1 | 81.2 | 70.9 | 65.6 |
| No merge | 79.7 | 71.3 | 66.9 | 82.1 | 71.5 | 66.8 | 47.1 | 61.7 | 42.6 |
| No split loss | 28.6 | 44.5 | 25.3 | 77.6 | 69.9 | 63.3 | 81.7 | 71.1 | 66.4 |
| Full method | 82.4 | 72.2 | 67.9 | 82.4 | 71.7 | 67.4 | 82.7 | 72.3 | 67.7 |

clusters is less than the ground truth, *i.e.*, $K_0=3$. If no split and merge scheme exists, the clustering algorithm can only achieve a 28.7% accuracy since the number of clusters remains unchanged during training. If there is no split phase, the performance of the proposed method is still bad. The proposed method can achieve a 79.7% accuracy when involving the split module, which indicates that the split phase is useful when the number of clusters is relatively small. Moreover, the split loss is essential to split the two sub-clusters. Otherwise, the proposed method can only achieve a 28.6% accuracy even if the split phase is included. When the full method is adopted, the proposed method achieves the best performance (82.4% accuracy).

We then evaluate the situation that the initial number of clusters is greater than the ground truth, *i.e.*, $K_0=20$. Similarly, if the merge phase is not applied, the proposed method cannot achieve satisfying performance since close clusters cannot be merged. By utilizing the merge and split phase, the proposed method achieves an 82.7% accuracy, which suggests that each component is essential.

**Running Times.** We compare the running time of our method with different initial $K$ to the original SCAN. Results show that the running time has varying degrees of increase, *e.g.*, on CIFAR-10, when the initial $K = 3$, the running time is 3× compared to a single run of SCAN. However, it is still much more efficient than training SCAN multiple times with a different K for model selection. Thus, the value and impact of our proposed module are clear.

### 4.4 Visualization Results.

We conduct visualization on the CIFAR-10 dataset to investigate the training process of the proposed method. Figure 2 shows the t-sne visualization of the proposed method during training, where different colors denote data points from different categories. The initial number of clusters is set to 3. At the beginning of training, the number of clusters is much smaller than the ground ruth. Therefore, the data points can hardly be divided in both feature and output spaces. During the training process, the proposed method gradually adjusts the number of clusters by splitting the clusters. As a result, the network trained by the proposed method converges to $k=10$, and the data points are classified in different clusters, as shown in the figure, resulting in the best accuracy (82.5%).

## 5 Conclusion

Deep clustering algorithms have achieved great success in unsupervised learning. However, most of these algorithms are parametric, which requires the number of clusters. When the number of clusters is unknown, the performance of existing deep clustering algorithms suffers a lot. To this end, we propose a plug-and-play module that can be directly applied to existing parametric clustering methods by analyzing the JS divergence

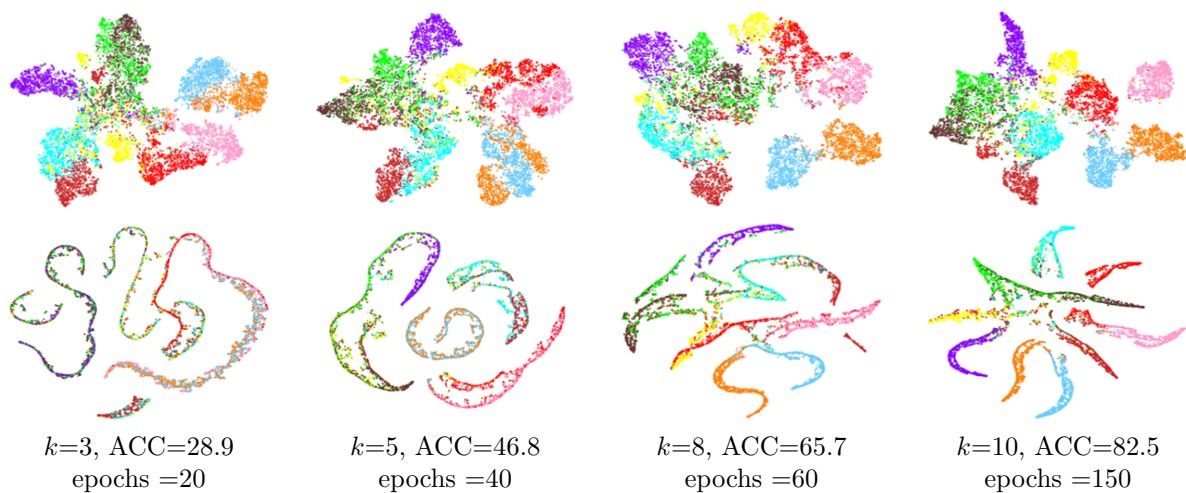

| $k$=3, ACC=28.9 | $k$=5, ACC=46.8 | $k$=8, ACC=65.7 | $k$=10, ACC=82.5 |
| epochs =20 | epochs =40 | epochs =60 | epochs =150 |

Figure 2: The training process of the proposed method. The images in the top row and the bottom row are the t-sne visualization results of features and outputs, respectively.

between clusters. Specifically, a split scheme is introduced to minimize the distance between data points in the same clusters. In contrast, a merge scheme is introduced to maximize the distance between data points in different clusters. As a result, the proposed method can successfully adjust the number of clusters when added to the existing clustering algorithms, achieving the state-of-the-art performance without knowing the number of clusters. Visualization and ablation studies demonstrate the effectiveness of the proposed algorithm in finding the accurate number of clusters.

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

# A   Appendix

**Proof of Proposition 1.**

*Proof.* For simplicity, we suppose cluster $K$ is split into two sub-clusters $K^1$ and $K^2$, the optimization problem becomes:

$$\min \sum_{k \in \{[K-1], K^1, K^2\}} \mathrm{D}(k,k) - \frac{\lambda}{K+1} \sum_{k_1, k_2 \in \{[K-1], K^1, K^2\}}^{k_1 \neq k_2} \mathrm{D}(k_1, k_2), \tag{15}$$

If the optimal value of problem 15 is smaller than that before split, the split phase should be applied. Therefore, the split condition can be formulated as:

$$\sum_{k \in \{[K-1], K^1, K^2\}} \mathrm{D}(k,k) - \frac{\lambda}{K+1} \sum_{k_1, k_2 \in \{[K-1], K^1, K^2\}}^{k_1 \neq k_2} \mathrm{D}(k_1, k_2) \leq \sum_{k \in [K]} \mathrm{D}(k,k) - \frac{\lambda}{K} \sum_{k_1, k_2 \in [K]} \mathrm{D}(k_1, k_2). \tag{16}$$

We can combine the same part in equation 16 and obtain:

$$\sum_{k \in \{K^1, K^2\}} \mathrm{D}(k,k) - \frac{2\lambda}{K+1} \mathrm{D}(K^1, K^2) \leq \mathrm{D}(K,K) - \left( \frac{\lambda}{K} - \frac{\lambda}{K+1} \right) \sum_{k_1, k_2 \in [K]}^{k_1 \neq k_2} \mathrm{D}(k_1, k_2). \tag{17}$$

Since $K^1$ and $K^2$ are split from $K$, we have:

$$D(K,K) = \sum_{k \in \{K^1, K^2\}} \mathrm{D}(k,k) + D(K^1, K^2), \tag{18}$$

and for any $k$,

$$D(k,K) = D(k,K^1) + D(k,K^2). \tag{19}$$

Further, we can conclude the following result:

$$\left( \frac{2\lambda}{K+1} + 2 \right) \mathrm{D}(K^1, K^2) \geq \frac{\lambda}{K(K+1)} \sum_{k_1, k_2 \in [K]}^{k_1 \neq k_2} \mathrm{D}(k_1, k_2). \tag{20}$$

$\square$

**Proof of Proposition 2.**

*Proof.* Similar with the class split phase, we can determine the threshold by analyzing the optimization problem:

$$\min \sum_{k \in \{[K-2], K^m\}} \mathrm{D}(k, k) - \frac{\lambda}{K-1} \sum_{k_1, k_2 \in \{[K-2], K^m\}}^{k_1 \neq k_2} \mathrm{D}(k_1, k_2), \tag{21}$$

where $K^m$ is the merged cluster. The merge condition can be formulated as:

$$\sum_{k \in \{[K-2], K^m\}} \mathrm{D}(k, k) - \frac{\lambda}{K-1} \sum_{k_1, k_2 \in \{[K-2], K^m\}}^{k_1 \neq k_2} \mathrm{D}(k_1, k_2) \leq \sum_{k \in [K]} \mathrm{D}(k, k) - \frac{\lambda}{K} \sum_{k_1, k_2 \in [K]}^{k_1 \neq k_2} \mathrm{D}(k_1, k_2). \tag{22}$$

Following the same strategy with equation 16 and 17, we can obtain:

$$\mathrm{D}(K^m, K^m) - \left(\frac{\lambda}{K-1} - \frac{\lambda}{K}\right) \sum_{k_1, k_2 \in \{[K-2], K^m\}}^{k_1 \neq k_2} \mathrm{D}(k_1, k_2) \leq \sum_{k \in \{K-1, K\}} \mathrm{D}(k, k) - \frac{2\lambda}{K} \mathrm{D}(K-1, K). \tag{23}$$

Then, we can also conclude as follows:

$$\left(\frac{2\lambda}{K} + 2\right) \mathrm{D}(K-1, K) \leq \frac{\lambda}{K(K-1)} \sum_{k_1, k_2 \in \{[K-2], K^m\}}^{k_1 \neq k_2} \mathrm{D}(k_1, k_2). \tag{24}$$

$\square$

**Proof of Lemma 3.**

*Proof.* Suppose that there are two clusters $K^1$ and $K^2$ satisfy both the merge and split condition, then we have:

$$\left(\frac{2\lambda}{K} + 2\right) \mathrm{D}(K^1, K^2) \leq \frac{\lambda}{K(K-1)} \sum_{k_1, k_2 \in \{[K-2]\}}^{k_1 \neq k_2} \mathrm{D}(k_1, k_2), \tag{25}$$

and

$$\left(\frac{2\lambda}{K+1} + 2\right) \mathrm{D}(K^1, K^2) \geq \frac{\lambda}{K(K+1)} \sum_{k_1, k_2 \in [K]}^{k_1 \neq k_2} \mathrm{D}(k_1, k_2). \tag{26}$$

which means:

$$\frac{\lambda}{2(K-1)(\lambda + K)} \sum_{k_1, k_2 \in \{[K-2], K^m\}}^{k_1 \neq k_2} \mathrm{D}(k_1, k_2) \geq \mathrm{D}(K^1, K^2) \geq \frac{\lambda}{2K(\lambda + K + 1)} \sum_{k_1, k_2 \in [K]}^{k_1 \neq k_2} \mathrm{D}(k_1, k_2). \tag{27}$$

The inequation can be simplified as:

$$\frac{\lambda}{2(K-1)(\lambda + K)} \sum_{k_1, k_2 \in \{[K-2], K^m\}}^{k_1 \neq k_2} \mathrm{D}(k_1, k_2) \geq \frac{\lambda}{2K(\lambda + K + 1)} \sum_{k_1, k_2 \in [K]}^{k_1 \neq k_2} \mathrm{D}(k_1, k_2)$$

$$\frac{1}{(K-1)(\lambda + K)} \left[\sum_{k_1, k_2 \in [K]}^{k_1 \neq k_2} \mathrm{D}(k_1, k_2) - 2\mathrm{D}(K^1, K^2)\right] \geq \frac{1}{K(\lambda + K + 1)} \sum_{k_1, k_2 \in [K]}^{k_1 \neq k_2} \mathrm{D}(k_1, k_2)$$

$$\left[1 - \frac{(K-1)(\lambda + K)}{K(\lambda + K + 1)}\right] \sum_{k_1, k_2 \in [K]}^{k_1 \neq k_2} \mathrm{D}(k_1, k_2) \geq 2\mathrm{D}(K^1, K^2) \tag{28}$$

$$\frac{\lambda - 2K}{2K(\lambda + K + 1)} \sum_{k_1, k_2 \in [K]}^{k_1 \neq k_2} \mathrm{D}(k_1, k_2) \geq \mathrm{D}(K^1, K^2)$$

Therefore, we have:

$$\frac{\lambda - 2K}{2K(\lambda + K + 1)} \sum_{\substack{k_1, k_2 \in [K] \\ }}^{k_1 \neq k_2} \mathrm{D}(k_1, k_2) \geq \mathrm{D}(K^1, K^2) \geq \frac{\lambda}{2K(\lambda + K + 1)} \sum_{\substack{k_1, k_2 \in [K] \\ }}^{k_1 \neq k_2} \mathrm{D}(k_1, k_2), \tag{29}$$

which raises a contradiction. Therefore, the split and merge condition cannot be met simultaneously. $\qquad\square$

**Proof of Proposition 5.**

*Proof.* From the proof of lemma 4, since the the number of clusters can only be monotonically changed and bounded ($K \geq 1$ and $K \leq m$ where $m$ is the number of samples), the proposed algorithm can finally converge.

Moreover, we conclude that the algorithm converged when $V(K^* + 1) > V(K^*) < V(K^* - 1)$, which means

$$K^* = \arg\min_K \sum_{k \in [K]} D(k, k) - \frac{\lambda}{K} \sum_{\substack{k_1, k_2 \in [K] \\ }}^{k_1 \neq k_2} D(k_1, k_2). \tag{30}$$

For the convergence speed, as shown in algorithm 1, the number of clusters can be changed up to about twice (each two classes satisfy the merge condition and merge into one class or each class satisfies the split condition and be split into two classes). Therefore, the proposed algorithm will converge at an $O(\frac{1}{2^n})$ speed, where $n$ is the number of acting the split-merge schemes.

$\qquad\square$

