# OpenReview forum: "Deep Plug-and-Play Clustering with Unknown Number of Clusters"
_TMLR — Accepted by TMLR_

### Review · Reviewer_zo9S · 2022-12-12

**Summary Of Contributions:**

This submission provides a clustering method that automatically adjusts the number of clusters. The method applies a clustering algorithm to the dataset, and then examines if the found clusters should be split or merged under certain criteria. The method can be combined with any existing clustering methods, and thus the title includes “plug and play”.



**Audience:**

Yes

**Claims And Evidence:**

No

**Requested Changes:**

- analyzing the convergence properties of the algorithm

- specify the assumptions needed for the algorithm to succeed (i.e., recovering the ground truth K)

- discussing (or analyzing) how the sample complexity would affect the performance

- modify the title and remove "deep clustering"

- discuss the relation between this method and existing split-and-merge methods in the literature

- discuss the impact of different D(.) functions when combining with different clustering criteria

- To make the claims/goals of the paper clearer. For example, "Instead of developing an effective non-parametric method, we aim to introduce a plug-and-play module that can be easily embedded into the state-of-the-art parametric clustering method without
affecting its performance." This is a little vague as it could be perceived as that the method can be embedded with many clustering methods. But the experiments were done by just combining the method with SCAN (Van Gansbeke et al., 2020). If that is the case, the method seems to be tuned for SCAN, but not "plug and play" for other clustering methods. In addition, it is debatable if SCAN is "the state-of-the-art".

**Strengths And Weaknesses:**

Strength

The paper deals with a hyper-parameter selection problem in clustering. This is essentially a “model order selection” problem in the literature, and is known to be a hard problem even if the mixtures are linear. The proposed approach is intuitive, and works to a certain extent.
The experiments show that the algorithm does estimate K well, given different initialization. This shows the robustness to the initial guess, which is an important trait of such algorithms.
The method is flexible, as it can be combined with any clustering methods as an additional module.

Weakness

The theoretical support for the proposed method is relatively weak. For example, proposition 1 is used to justify the splitting criterion. But proposition 1 only states that under certain conditions the cost value of (1) will decrease. However, this seems to be an indirect measure of success as (1) is also affected by a hyper-parameter lambda. Decreasing (1) does not necessarily mean “good” clustering structure found.

There are some points that were not crystal clear. The clarity of the technical assumptions may be improved. For example, the choice of D(.) seems to be a bit casual. This work uses the JS divergence, using “symmetry and easy to calculate” as the motivation. But does this implicitly assume that the support of K^1 and K^2 are the same? There is another assumption that x are iid as distribution comparison is used as a criterion. It is unclear how the non-iid case would affect the performance.

The depth of the submission may not be satisfactory. The sample complexity is not discussed, but the JS-divergence can only be approximated by samples.

As an algorithm, the convergence issue was not discussed. Will the method ever converge, or is there a risk of diverge? There is a lack of discussion.

I found that the title of the submission may not exactly reflect the contribution. The title implies that this is a ``deep plug and play’’ work, but the method actually can be combined with any clustering method. The method does not really exploit any “deep learning” structure or property. Having “deep” in the title is a little misleading.

At a higher level, the method may be problematic when combining with some clustering methods. The criterion (1) uses its own definition of clusters (e.g., maximizing intra-cluster distance while minimizing inter-class distance). This was fleshed out using a distribution “distance” metric. But plenty of clustering methods do not use the “distance” metric in (1). For example, k-means uses Euclidean distance, and subspace clustering uses angles between susbpaces. It is understandable that the D(.) function in (1) can be replaced by other divergence measures, but how they will work in practice is unclear and not supported by analysis or numerical evidence.

Finally, split-and-merge using the 2-cluster clustering method (like in line 6 of algorithm 1) as a subroutine to check if a cluster needs to be split is a long-existing idea; see Chaudhuri, Debasis, B. B. Chaudhuri, and C. A. Murthy. "A new split-and-merge clustering technique." Pattern Recognition Letters 13.6 (1992): 399-409.

---

> ### Author Response · Authors · 2023-01-12
> **Rebuttals by Paper522 Authors**
>
> We thank the reviewer for the detailed comments and suggestions. Bellow, we provide detailed responses to the weaknesses and questions.
>
> ***Q: The theoretical support for the proposed … not necessarily mean “good” clustering structure found.***
>
> ***A:*** It’s clear that when K is closer to the ground-truth class number, the lower the optimization cost for formulation (2), i.e.
> \begin{equation}
> \mbox{arg}\min_K \sum_{k\in [K]} D(k,k) - \frac{\lambda}{K} \sum_{k_1,k_2\in [K]}^{k_1\neq k_2}D(k_1,k_2),
> \end{equation}
> will be. The split condition, i.e. proposition 1 or merge condition, i.e. proposition 2 are deduced by assuring the overall optimization cost degrades after split or merge. We show the detailed deduction for the above 2 proposition. Besides, $\lambda$ is only a hyper-parameter to balance the Compactness and Separation term, which does not affect the optimization problem.
>
> ***Q: Discuss choice and the impact of different D(.) functions***
>
> ***A:*** Thanks for the suggestion. We use JS-divergence as the D(.) metrics because the output of a sample in neural networks can be naturally viewed as a probability distribution. Therefore, it is direct and proper to use divergences as a distance metric to measure the cluster similarities. We also conduct a simple experiment on CIFAR-10 dataset to explore different D(.). Resutls show that our proposed method can be effective with different D(.) that can reflect the spatial distance, except for the D(.) that mainly reflects the directions. e.g cosine similarity.
> |  |  |  |  |  |
> |---|---|---|---|---|
> | **D(.)** | JS-Divergence | L1 distance | Euclidean distance | Cosine Similarity |
> | **Inferred K** | 10.0±0.0 | 10.5±0.5 | 10.5±0.5 | 8±2 |
> | **ACC** | 82.6±0.3 | 77.8±1.5 | 81.3±1.8 | 70.8±3.5 |
>
> ***Q: Discussing how the sample complexity would affect the performance***
>
> ***A:*** Thanks for the nice suggestion. We conduct an experiment on CIFAR-10 datasets (50,000 for training and 10,000 for validation) with different amount of training data to disscuss the sample complextiy.
> |  |  |  |  |  |  |
> |---|---|---|---|---|---|
> | **Sample Ratio** | 1 | 0.5 | 0.2 | 0.05 | 0.01 |
> | **Inferred K** | 10.0±0.0 | 10.0±0.0 | 10.0±0.0 | 9.7±2.3 | 5.8±0.4 |
> | **ACC** | 82.6±0.3 | 82.5±0.3 | 82.0±0.5 | 71.1±4.5 | 48.2±4.8 |
>
> We observe that when the training amount of data reduces to 1/5,  our proposed method can still acheive stable and satisfying clustering results.  Further reducing the data amount will cause damage to the clustering performance. However, one of the important reason is that the clustering method is based on deep neural networks, which need enough data to learn good representations. We directly run SCAN given K=10 on 5% training data, the final ACC is only 75.2, which is comparable to our results. Therefore, we think that the effect of sample complexity to our proposed method is nearly the same as its effect on the original parametric clustering methods.
>
> ***Q:  Analyzing the convergence properties of the algorithm and specifying the assumptions needed for the algorithm to succeed (i.e., recovering the ground truth K).***
>
> ***A:*** Thanks for the problem. We add the analysis and prove the convergence of the proposed method in the revised version.
>
> ***Q: To make the claims/goals of the paper clearer, ..., for other clustering methods.***
>
> ***A:*** We have done experiments to show the "plug-and-play" feature of our methods. Table 4 shows when embedded with our proposed method, other deep parametric clustering frameworks, e.g. NNM GCC, can automatically ajust the class number and acheive satisfying clustering results, given an inaccurate initial K.
>
> Besides, our proposed methods also can generalize to other domains of data. We  further embed our module into a deep clustering method IDEC (Xifeng Guo et al., 2017). and run an experiment of text clustering task on Reuters dataset with 4 clusters. The results show that our method can be "plug-and-play" and generalize well to other domains.
> |  |  |  |  |  |  |
> |---|---|---|---|---|---|
> | **Reuters-10K (K=4)** |  |  |  |  |  |
> | **Initial K** | 2 | 4 | 10 | 20 | 40 |
> | **IDEC** | 59.5 | 71.1 | 44.2 | 28.8 | 15.4 |
> | **IDEC+ours** | 70.8 | 71.3 | 76.2 | 73.6 | 72.6 |

---

> > ### Comment · Reviewer_zo9S · 2023-01-30
> > **thanks for the reply**
> >
> > I would like to thank the authors for the replies and additional experiments.
> >
> > My further comments are as follows:
> >
> > The cost function in (2): In the reply, the authors said that lambda does not affect the optimization problem. This is a bit counter intuitive. Will setting lambda to close to zero or close to +infity having the same effect on finding the ground-truth K? It is hard to digest this statement.
> >
> > The choice of D(,). The authors replied that “We use JS-divergence as the D(.) metrics because the output of a sample in neural networks can be naturally viewed as a probability distribution. Therefore, it is direct and proper to use divergences as a distance metric to measure the cluster similarities.” This seems to be not enough to support the choice of JS, as other divergence measures exist (e.g., KL divergence and Wasserstein distance) for measuring the similarity of distributions.
> >
> > Sample complexity:  I appreciate that the authors added one table to show how sample complexity affects the performance. But this simple table does not show how other methods perform. Are the baselines less sensitive to reducing the training samples? This is unknown. More importantly, how does the sample complexity interplay with the neural network’s complexity, as the paper positions the method as a “deep clustering” work? There is a lack of understanding to these aspects.
> >
> > Convergence: It is unclear to me what is the convergence criterion used in the added proposition 3. The proposition states that there is no K that can satisfy the splitting and merging criteria simultaneously. However, this does not say the algorithm would stop at the correct K (or would stop at any K). It also does not say anything about how fast the algorithm would find the correct K. It does not seem to be an ordinary convergence statement.
> >
> > Regarding “deep clustering” in the title: The authors argued that their split-and-merge would change the network architecture in each iteration. For this reason, I agree to call the method “deep clustering”.

---

> ### Author Response · Authors · 2023-01-13
> **Rebuttals by Paper522 Authors**
>
>
> ***Q: Modify the title and remove "deep clustering" and having “deep” in the title is a little misleading.***
>
> ***A:*** Sorry for the confusion. We admit that the split-and-merge strategy is a long-existing idea. However, our work is the first one to apply this strategy to deep parametric clustering framework. First, we use the output of the neural network, i.e logits as the input for calculating cluster similarities, which is simple yet effective. Second, we introduce assistant 2-class networks in the split stage to decide the split operation and introduce a split loss in training. Third, we change the architecture of the network after split-and-merge stage by merging or splitting the corresponding weights. So overall, our method is specially designed for deep clustering frameworks, whose output exactly represent the clustering results.
>
> ***Q: Discuss the relation between this method and existing split-and-merge methods in the literature.***
>
> ***A:*** Current clustering frameworks that use split-and-merge strategy are mainly developed for non-deep models and do not consider the way to change the clustering model architecture, e.g. "Cluster merging and splitting in hierarchical clustering algorithms" or "Automatic Cluster Number Selection using a Split and Merge K-Means Approach".  Besides, these algorithms are not flexible, for example, a strategy designed for K-means ususally cannot be easily applied to other methods, while our proposed module can be flexibly embedded to any state-of-the-art deep parametric clustering frameworks.  The novelty of our work lies in the specific split-and-merge design for deep clustering neural networks, and adaptively obtaining the thresholds for split or merge by analyzing the optimization problem. Experiments also demonstrate that equipped with our proposed module, deep clustering networks not only can achieve better performances, but efficient and easy to use in real-world scenarios where class numbers are often unknown.

---

### Review · Reviewer_r4rJ · 2022-12-29

**Summary Of Contributions:**

This paper proposes a clustering strategy to group data without knowing the class number.

**Audience:**

Yes

**Claims And Evidence:**

No

**Requested Changes:**

Please see Section *Strengths And Weaknesses

**Strengths And Weaknesses:**

The paper is easy to follow. But some drawbacks and suggestions are as follows:

1. The authors claim "current number of clusters K' is smaller than the optimal value K^∗." in the first paragraph of section 3.1. This is one of the basic assumptions of this paper. Are there any observations or proofs to support this?

2. In section 3.1, "the optimization problem 2" is often used but what does it refer to?

3. The authors use "plug" to describe their contribution, but "framework" would be more appropriate. "plug" often describe a relatively independent module rather than a learning strategy.

4. How to guarantee the proposed strategy (Alg. 1) is convergent. There is no theoretical proof, and this is also not obvious.


5. In table 1, how the NMI, ACC and ARI are computed. To my knowledge, the computation requires the ground-truth label, which means the proposed method groups the data with the right class number (e.g. 10 classes for CIFAR10, 100 classes for CIFAR100). But apparently, this is not always the case.

6. How to explain the hyper-parameter \lambda, since the authors want to eliminate the widely-used hyper-parameter, i.e. class number, but introduce another one. In Table 6, we can see the performance is not stable when setting the parameter \lambda with different values.

7. In table 5, the proposed method is initialized with different K. Does it separate the data into groups with a same class number? If not, how to compare their performance.

8. Another popular metric, Purity, could be considered in the experiments.

9. In the tables, there are lots of "-". what do they mean?

---

> ### Author Response · Authors · 2023-01-12
> **Rebuttals by Paper522 Authors**
>
> We thank the reviewer for the detailed comments and suggestions. Bellow, we provide detailed responses to the weaknesses and questions.
>
> ***Q1: About basic assumptions "current number of clusters K' is smaller than the optimal value K^∗"***
>
> ***A:*** Sorry for the confusion. We don’t have this basic assumption. "current number of clusters K' is smaller than the optimal value K^∗" is a one of the situation where we need to execute the cluster split operation. In section 3.2, we discuss another situation where current K' is larger than the optimal values K^*, and in which we need to execute the cluster merge operation. In our experiment, for example, on CIFAR-10 dataset, our method can always infer the optimal K*=10 no matter the initial K’=3 or K’=20.
>
> ***Q2: What dose "the optimization problem 2" refer to?***
>
> ***A:*** Sorry, it's a typo. It should be "the optimization problem (2)", which refers to the optimization problem to clustering with unknown number of clusters.
>
> ***Q3: Use "Framework" instead of "Plug"***
>
> ***A:*** Thanks for the suggestion. We use plug-and-play to highlight the transferability of our method. Our method is an independent split-and-merge module, which can be combined with different kinds of parametric deep clustering frameworks, e.g. SCAN, GCC, NNM, etc.
>
> ***Q4: Guarantee the proposed strategy (Alg. 1) is convergent"***
>
> ***A:*** Thanks for the problem. We add the analysis and prove the convergence of the proposed method in the revised version.
>
>
> ***Q5: How the NMI, ACC and ARI are computed***
>
> ***A:*** As you mentioned, the computation of NMI, ACC and ARI requires the ground-truth label. In our experiment, we follow the conventional clustering settings, which often run experiments on datasets with ground-truth lables. However, labels are only used for evaluating the performance of proposed algorithms and the training of clustering algorithms is completely unsupervised, which is a common case in clustering scenarios.
>
> ***Q6: How to explain the hyper-parameter $\lambda$***
>
> ***A:*** $\lambda$ is a hyper-parameter to balance the intra-class similarity and inter-class similarity in the optimization problem. Compared to the original class number K, which needs to be tuned for each dataset, our proposed $\lambda$ is more robust. In Table 6, we can see when \$lambda$ is within a reasonable scope, the final clustering results won’t change a lot. Besides, we can see $\lambda$ is robust for different datasets with different class numbers. So in our paper, we set $\lambda=2$ for all datasets, which avoids the heavy hyper-parameters' tuning in different datasets.
>
> ***Q7:  In table 5, the proposed method ... a same class number?***
>
> ***A:*** The goal of this ablation study is to show our proposed module can help the deep parametric clustering framework, e.g. SCAN etc. to achieve satisfying results when the initial class number K is far from the ground truth. Note that deep parametric clustering methods cannot adjust the class number during training, so when given an incorrect class number K, these method will fail, i.e. ACC degradation. But when equipped with our module, these methods can correctly and automatically infer the optimal K for different initial K', which makes these methods applicable in real-world scenarios.
>
> ***Q8:  Considering purity metrics***
>
> ***A:*** Thanks for the suggestion. We mainly applied our propose module in deep parametric clustering frameworks, e.g. SCAN, NNM, GCC etc., therefore for fair comparison, we just follow the experimental settings in the previous papers, which commonly use ACC, ARI and NMI as the evaluation metrics.  Besides, we observe that the computation of purity is smililar to ACC, except that ACC computation need an extra hungarian match process to align the predictions to ground-truth labels. We measure the purity metrics on CIFAR-10 dataset and find that when the optimal K is reached, the value of purity equals to ACC.
>
> ***Q9: In the tables, there are lots of "-". what do they mean?***
>
> ***A:*** Sorry for the confusion. “-“ means lack of value.

---

### Review · Reviewer_TuEe · 2022-12-30

**Summary Of Contributions:**

The paper proposes a simple split-and-merge schema for finding the correct number of clusters for any deep clustering approach that requires the number of clusters as input. At its core, the proposed method is similar to hierarchical clustering algorithms (like BIRCH) that split and merge clusters to minimize the within-cluster distance and maximize the between-cluster distances. In short, after training the deep clustering algorithm of interest with a predefined number of clusters $K'$, the authors carry out a cluster split step based on the Jensen-Shannon (JS) divergence between two possible subclusters and obtain a new $K$. Then, they retrain (or possibly fine-tune) the clustering network based on the new $K$, and enter a merge phase, where different clusters are merged to minimize the within-cluster and maximize the between-cluster distances. The process is repeated until convergence. The authors apply their plug-and-play split-and-merge algorithm to SCAN and show improvement in clustering performance compared to various baselines on CIFAR-10, CIFAR-100, STL-10, and ImageNet-10 datasets.

Unfortunately, too many critical details are currently missing from the paper.

**Audience:**

Yes

**Claims And Evidence:**

No

**Requested Changes:**

Please refer to the weaknesses enumerated above.

**Strengths And Weaknesses:**

## Strengths:

* The plug-and-play nature of the proposed framework is interesting
* The paper is very well aligned with classic hierarchical clustering approaches like BIRCH

## Weaknesses:

* Critical - Too many crucial details are missing from the paper
  * How is the JS divergence calculated on the empirical measures? The authors briefly mention that they use binary feed-forward classifiers to measure the JS divergence. First, this strategy requires training a binary classifier to evaluate the JS divergence between **any** two empirical distributions, which raises a question about the computational efficiency of the proposed approach. Second, training such binary classifiers will only approximate the JS divergence.
  * When it comes to splitting a cluster, the subclusters $K_1$ and $K_2$ are unknown. An optimization is required over $K_1$ and $K_2$ to maximize the JS divergence between the subclusters. Also, given that calculating the JS divergence requires training a binary classifier, it needs to be clarified how this optimization is carried out.
  * It is unclear why the proposed schema should converge! There is no clarification regarding the convergence of the proposed framework.
  * Is the network trained from scratch on the new $K$ after split or merge, or is it fine-tuned?

* False Promise - The approach is motivated by saying that existing approaches that dynamically update $K$ are costly for deep clustering algorithms. However, the proposed approach is also very costly. In addition to requiring an initial deep clustering (baseline), the proposed algorithm iteratively updates $K$, where in each iteration, the deep clustering algorithm must be executed after the split and after the merge phases. Even not considering the computational complexity of the split and merge steps, each iteration requires running the deep clustering algorithm twice! To make matters worse, there is no guarantee that the method will converge in a few iterations.

* Writing - The paper can benefit from rewriting some sentences, and fixing some typos:
  * E.g., Page 4, Section 3.1, second line: "becomes the dominate term" should be "becomes the dominant term"
  * E.g., Page 4, Section 3.1, last line: "inequation 3" should be "inequality 3"

---

> ### Author Response · Authors · 2023-01-12
> **Rebuttals by Paper522  Authors**
>
> We thank the reviewer for detailed comment. Bellow, we provide detailed responses to the weaknesses and questions.
>
> ***Q1：How is the JS divergence calculated ... will only approximate the JS divergence.***
>
> ***A：*** For deep clustering algorithms, we don't need to use feed-forward classifiers to measure the JS-divergence, because the output of these algorithms has already represented the probability distribution of each sample. Therefore, JS-divergence can be computed naturally over these distributions. Specifically,  we will explain our methods based on the SCAN framework.
>
> First, the original deep clustering algorithm, e.g. SCAN, uses a deep neural network to separate samples into different clusters, where the output dimension of the network equals to the number of clusters. Take a dataset with N samples and M clusters for example, the output of the original algorithm is an N by M matrix, where each row represents a sample, each column represents a cluster and the value is the probability of a sample that belongs a specific cluster.
>
> Second, after obtaining the N by M output matrix, we use argmax to decide the clustering class of each sample, and then divide N samples into M clusters. In merge phase, for every two clusters, we calculate the JS divergence between samples in two clusters, because the output of the neural networks can be viewed as probability distribution. Without loss of generality, we take the first two clusters i, j as example. For a sample r, we can obtain the its probability belonging to cluster i and cluster j and get a probability vector [r_i, r_j] in the output matrix. After getting the probability vectors of samples in cluster i and cluster j, we calculate the JS divergence between any two samples, and sum up all the JS divergence of samples to get the overall JS divergence between any two clusters.
>
> ***Q2: When it comes to splitting a cluster ... this optimization is carried out.***
>
> ***A:*** In split phase, considering current network has M outputs, we directly add M assistant 2-class feed-forward neural networks to the M dimensional outputs of original network, and the overall number of clusters is expanded to 2M. Here, we still use the original deep clustering algorithm to perform the binary classification. Take the first cluster for example, the training of the 2-class network only invovles samples that belong to this cluster. So the overall training of M assistant networks only goes over the entrie datasets once. The optimization follows the original training algorithm, except we introduce a split loss to make the sub-clusters more distinguishable. After training the assistant 2-class classification head, we can further split the samples in this cluster into two sub-clusters. Following the process of JS-divergence computation, we can calculate the JS-divergence of these two sub-clusters rather than directly use binary classifiers to approximate it. If the split condition is satisfied, the corresponding neural network architecture will be changed, and the output dimension will expand to M+1.
>
> ***Q3: Guarantee the convergence of the proposed framework.***
>
> ***A:*** Thanks for the problem. We add the analysis and prove the convergence of the proposed method in the revised version.
>
> ***Q4: Computational efficiency and time***
>
> ***A:***Thank you for the nice concern. First, for the overall training time, it depends on how far it is from the initial K to the ground-truth K. We compare the running time of our method with different initial K to the original SCAN. The results show that the running time has varying degrees of increase. However, it is still much more efficient than training SCAN multiple times with a different K for model selection.
> |  |  |  |  |  |  |
> |---|---|---|---|---|---|
> | **K** | 3 | 20 | 40 | 50 | 100  |
> | **Time** | 3x | 2.6x | 4.5x | 6.9x | 11.9x  |
>
> Second, the training of binary clsassifiers in split stage between every two sub-clusters only uses data samples classified in the original cluster. So no matter how large the number of K is, the total amount of data used for one epoch training remains the same. We conduct an ablation study on 50,000 CIFAR images. By performing split stage with one epoch training with K varying from 3 to 100, we show that the training time will not change with the increase of K.
> |  |  |  |  |  |
> |---|---|---|---|---|
> | **K** | 3 | 20 | 50 | 100  |
> | **Split Training Time(s)**|53.94|55.31|55.45 | 55.65  |
>
> Therefore, the value and promise of our proposed module are clear.
>
> ***Q5: Trained from scratch or fine-tuned?***
>
> ***A:*** It is fine-tuned. We start from an initial K’ and gradually adjust to infer the optimal K. During the process, the adjustment of number of K and the training of the network are executed simultaneously. So when the optimal K is found, the network converges as well, and so the clustering is completed.
>
> ***Q6: Writings and typos***
>
> ***A:*** Thanks for the suggestions. We will revise them.

---

### Decision · Action_Editors · 2023-05-01

**Recommendation:** Accept with minor revision

**Comment:**

The paper studies a simple split-and-merge schema for finding the correct number of clusters for any deep clustering approach, which requires the number of clusters as input. At its core, the proposed method is similar to hierarchical clustering algorithms that split and merge clusters to minimize the within-cluster distance and maximize the between-cluster distances. In a nutshell, after training the deep clustering algorithm of interest with a predefined number of clusters, the authors carry out a cluster split step based on the Jensen-Shannon (JS) divergence between two possible subclusters and obtain a new. Then, they retrain the clustering network and enter a merge phase, where different clusters are merged to minimize the within-cluster and maximize the between-cluster distances. The authors apply their plug-and-play split-and-merge algorithm to SCAN and show improvement in clustering performance compared to various baselines on CIFAR-10, CIFAR-100, STL-10, and ImageNet-10 datasets.

The general plug-and-play idea could lead more papers to focus on the deep clustering problem, which is good for the area. Specifically, the plug-and-play nature of the proposed framework is very interesting, and the paper is very well aligned with classic hierarchical clustering approaches like BIRCH. However, there still exists some potential issues (see https://openreview.net/forum?id=6rbcq0qacA&noteId=vmXRjmIzfU) proposed by reviewer zo9S, e.g., convergence theorem (Proposition 5), which should be further revised and fixed (see https://openreview.net/forum?id=6rbcq0qacA&noteId=bI1RVoQZjE). In general, the main idea of this paper is interesting and reasonable. The authors address most of reviewers' concerns in their rebuttal. Thus, I would like to recommend accept with minor revision.

**Audience:**

Yes

**Claims And Evidence:**

Yes